# Genome-Wide Identification and Expression Analysis of Auxin Response Factor Gene Family in *Linum usitatissimum*

**DOI:** 10.3390/ijms241311006

**Published:** 2023-07-02

**Authors:** Yanni Qi, Limin Wang, Wenjuan Li, Zhao Dang, Yaping Xie, Wei Zhao, Lirong Zhao, Wen Li, Chenxi Yang, Chenmeng Xu, Jianping Zhang

**Affiliations:** Institute of Crop, Gansu Academy of Agricultural Sciences, Lanzhou 730070, China; qiyanni@gsagr.cn (Y.Q.); wanglm@gsagr.cn (L.W.); liwenjuan@gsagr.ac.cn (W.L.); dangzhao@gsagr.cn (Z.D.); xieyp2016@gsagr.cn (Y.X.); zhaowei@gsagr.ac.cn (W.Z.); zlrgsau@163.com (L.Z.); lw3028149845@163.com (W.L.); ycx9925@163.com (C.Y.); 18735906104@163.com (C.X.)

**Keywords:** flax, auxin response factor (ARF), bioinformatics, abiotic stress, expression analysis

## Abstract

Auxin response factors (ARFs) are critical components of the auxin signaling pathway, and are involved in diverse plant biological processes. However, *ARF* genes have not been investigated in flax (*Linum usitatissimum* L.), an important oilseed and fiber crop. In this study, we comprehensively analyzed the *ARF* gene family and identified 33 *LuARF* genes unevenly distributed on the 13 chromosomes of Longya-10, an oil-use flax variety. Detailed analysis revealed wide variation among the ARF family members and predicted nuclear localization for all proteins. Nineteen LuARFs contained a complete ARF structure, including DBD, MR, and CTD, whereas the other fourteen lacked the CTD. Phylogenetic analysis grouped the LuARFs into four (I–V) clades. Combined with sequence analysis, the LuARFs from the same clade showed structural conservation, implying functional redundancy. Duplication analysis identified twenty-seven whole-genome-duplicated *LuARF* genes and four tandem-duplicated *LuARF* genes. These duplicated gene pairs’ K_a_/K_s_ ratios suggested a strong purifying selection pressure on the *LuARF* genes. Collinearity analysis revealed that about half of the *LuARF* genes had homologs in other species, indicating a relatively conserved nature of the *ARFs*. The promoter analysis identified numerous hormone- and stress-related elements, and the qRT-PCR experiment revealed that all *LuARF* genes were responsive to phytohormone (IAA, GA3, and NAA) and stress (PEG, NaCl, cold, and heat) treatments. Finally, expression profiling of *LuARF* genes in different tissues by qRT-PCR indicated their specific functions in stem or capsule growth. Thus, our findings suggest the potential functions of *LuARFs* in flax growth and response to an exogenous stimulus, providing a basis for further functional studies on these genes.

## 1. Introduction

Auxins are a class of phytohormones found extensively in plants. They play vital roles in regulating various plant processes, including seed germination, root architecture and development, shoot elongation, inflorescence development, xylem and phloem differentiation, fruit development, leaf expansion and senescence, shoot apical dominance, and plant stress response [1,2,3,4]. Research has demonstrated that the changes in auxin levels regulate the transient expression of auxin response genes, triggering the reprogramming of downstream genes and consequently influencing plant growth. The major auxin response genes include auxin/indole-3-acetic acid (*AUX/IAA*) family, auxin response factor (*ARF*) family, gretchenhagen 3 (*GH3*) family, and small auxin upregulated RNAs (*SAURs*) [5]. Among these, ARFs, the well-known transcription factors, are essential components of the auxin-mediated signaling pathways. They recognize and specifically bind with the auxin-response element (TGTCNN; AuxRE) in the promoter region of the target genes and regulate their transcription [6,7,8].

A typical ARF protein contains three conserved domains: an N-terminal DNA-binding B3 domain (DBD), a variable middle region (MR), and a C-terminal domain (CTD), which may form a type I/II Phox and Bem1p (PB1) protein–protein interaction domain [9,10,11]. The DBD of the ARF protein recognizes the AuxREs, while the MR acts as an activation domain (AD) or repression domain (RD). The amino acid composition of the MR determines whether ARFs activate or inhibit the transcription of downstream genes. Typically, glutamine (Q), serine (S), and leucine (L) are more in transcriptional activators, while serine (S), proline (P), glycine (G), and leucine (L) are abundant in transcriptional repressors [6,9]. However, CTD is present in only a few ARF proteins; it plays a vital role in plant-specific responses to auxin by mediating the interactions between Aux/IAA and ARF in a dose-dependent manner [12,13]. In plants, at low auxin levels or in the absence of auxin, the Aux/IAA proteins form multimers with ARFs; this complex binds to the co-repressor TOPLESS (TPL) and its family proteins (TPRs), inhibits the ARFs, and prevents the expression of downstream auxin-responsive genes [14]. However, when the auxin concentration is above a certain level, Aux/IAA is ubiquitinated by the ubiquitin ligase SCF^TIR1/AFB^ subunit (transport inhibitor response 1, TIR1), and subsequently, degraded through the 26S proteasome pathway [15,16], releasing the ARFs to regulate the transcription of downstream genes. 

Research has elucidated the functions of *ARF* genes in several plant species. In *Arabidopsis thaliana*, *AtARF1* and *AtARF2* regulate leaf senescence and floral abscission [17]; *AtARF3* regulates shoot apical meristem maintenance [18]; *AtARF5* regulates the patterning of vascular bundles, the development of hypocotyls, and the development of female and male gametophytes [19,20]; *ARF6* and *ARF8* regulate flower maturation [21]; and *AtARF7* and *AtARF19* regulate lateral root formation [22]. In rice, *OsARF1* and *OsARF12* play a vital role in regulating vegetative and reproductive development and phosphate homeostasis, respectively [23,24]. *GmARF8a* and *GmARF8b* of soybean influence nodulation and lateral root development [25], *GhARF2* and *GhARF18* of cotton regulate fiber cell initiation [26], and *ARF7* of poplar regulates cambial activity [27]. Overexpressing of *Magnolia sieboldii ARF5* in *Arabidopsis* has effects on vegetative and reproductive growth [28]. Silencing of *LsARF8a* in lettuce by the virus-induced gene silencing (VIGS) method indicated that *LsARF8a* conduced to the thermally induced bolting [29]. In addition, the *ARF* gene family of various plant species has been associated with responses to phytohormones and abiotic stresses [30,31,32]. For example, salicylic acid and heavy metals significantly upregulated *BdARF4* and *BdARF8* in *Brachypodium distachyon* [30]. The tomato knockout of *SlARF4* exhibited improved water deficit resistance [33]. In *Populus trichocarpa*, 15 *ARF* genes have been associated with abiotic stress and ABA response [34].

*ARF* genes have been extensively identified and characterized in many plants, including *Oryza sativa* [35], *Gossypium hirsutum* [26], *Setaria italica* [36], *Glycine max* [37], and *Magnolia sieboldii* [28]; however, no study has been carried out in flax. As one of the first domesticated crops, flax (*Linum usitatissimum* L.) is extensively cultivated in various countries for its oil and fiber [38]. It is a diploid and self-pollinated crop (2n = 30) with a small genome (306 Mb) and a short life cycle [39,40]. With the increasing demand for seed and fiber, the production and quality of flax need to be improved. Recently, the flax genome has been released, providing a basis for developing high-yielding and high-quality varieties through molecular breeding [40]. Considering the vital role of *ARF* gene members in regulating growth and development in various species, it is worth investigating the gene family in flax (*LuARF*). Therefore, the present study conducted a genome-wide analysis to identify the flax *ARF* gene family members. We analyzed their chromosomal distribution, gene structure, conserved domain, evolutionary relationship, and *cis*-acting elements. We further assessed their expression patterns in various tissues and under phytohormone and abiotic stress treatments to determine the potential functions. Our results will provide comprehensive information on flax *ARF* genes and lay a foundation for further functional studies on the *LuARF* gene family.

## 2. Results

### 2.1. Identification of ARF Genes in Flax

The known ARF protein sequences from nine species were used as queries for the BLASTP search to identify the flax sequences (Appendix A) [41]. Further, based on the presence and integrity of the conserved Auxin_resp and B3 domains in the Pfam and CDD databases, we identified 33 (named *LuARF1*–*LuARF33*), 35, and 34 *ARF* members in Longya-10, Heiya-14, and pale flax, respectively (Table 1 and Appendix A). Further analysis showed that the length of the ARF proteins varied from 470 (LuARF28) to 1123 (LuARF21) amino acids (aa), 401 to 1127 aa, 478 to 1058 aa in Longya-10, Heiya-14, and pale flax, respectively, while the predicted molecular weight (MW) ranged from 52.81 to 125.34 kDa, 44.04 to 125.79 kDa, and 52.85 to 117.28 kDa, and the isoelectric point (pI) varied from 5.45 to 8.59, 5.45 to 9.11, and 5.57 to 8.21, respectively. The subcellular localization analysis predicted that the ARF proteins in these three genomes were localized in the nucleus. All information, including gene name, locus ID, chromosome location, protein length, MW, pI, number of exons, and subcellular localization, of *LuARF* genes and the *ARFs* of Heiya-14 and pale flax are listed in Table 1 and Appendix A, respectively, and the ARF protein sequences are shown in Appendix A.

### 2.2. Sequence Analysis of Flax ARFs

Further, to assess the genetic relationships among the LuARFs, a phylogenetic tree was constructed using the LuARF protein sequences (Figure 1a). The analysis showed that the LuARFs could be divided into four groups (groups A to D) and most LuARF members existed in pairs. Further analysis of the gene structure by comparing the cDNA sequences with the genomic DNA sequences using Gene Structure Display Server (GSDS) revealed that the length of *LuARF* genes varied significantly, with *LuARF9* being the longest (26.0 kb) and *LuARF13* being the shortest (2.4 kb) (Figure 1b and Table 1) [42]. The *ARF* genes in Heiya-14 and pale flax were 1.4 to 14.6 kb and 2.2 to 14.6 kb long, respectively (Appendix A). These observations indicated significant differences among the members of the same cultivar. The number of exons was also remarkably different (2 to 14 or 15) among the *ARF* genes in the three genomes. Twenty-two *LuARF* genes (66.7%) had ≥10 exons, while seven had 2–5 exons, and the other four had 8–9 exons. The number of exons in the *ARF* genes of Heiya-14 and pale flax was similar to that of Longya-10.

Further analysis based on Pfam and CDD databases indicated that 19 out of 33 LuARF proteins had the typical ARF structure composed of a conserved DBD domain (B3 and AUX_RESP), a variable MR domain, and a CTD (Aux/IAA), while the remaining 14 lacked the CTD (Appendix A). Analysis of the conserved motifs in LuARF proteins using the Multiple Expectation maximization for Motif Elicitation (MEME) program identified 92 motifs, named motif 1 to motif 92 (Figure 1c and Appendix A). Motifs 1, 2, and 6, which form the B3 domain, existed in all LuARFs. Motifs 5, 7, 8, 12, and 14 corresponded to the ARF domain; of them, motif 8 existed in all LuARF proteins, while the rest only existed in some LuARF proteins. Meanwhile, motifs 4, 9, and 16 constituted the CTD. Motifs 3, 6, and 11 existed in more than 90.9% of the gene members, indicating their significantly conserved nature among the LuARF proteins. Seventy-two motifs were found in less than half of LuARF proteins, and twenty-seven motifs existed only in 6% LuARFs. These differences in the distribution of motifs indicated functional diversity among the LuARF members. Further, based on the MR amino acid composition and the presence or absence of the CTD, LuARF proteins were divided into three groups: proteins with a DBD, an activator MR enriched with Q, S, and L, and a CTD (LuARF6, 9, 10, 11, 15, 17, 20, 21, 22, 23, 26, 32, and 33); proteins with a DBD, a repressor MR enriched with S, L, P, and G, and a CTD (LuARF1, 7, 14, 16, 24, and 31); and proteins with a DBD and a repressor MR, but lacking a CTD (LuARF2, 3, 4, 5, 8, 12, 13, 18, 19, 25, 27, 28, 29, and 30) (Appendix A and Appendix A). 

### 2.3. Phylogenetic Analysis of ARF Proteins

Then, to assess the evolutionary significance of ARFs in flax domestication, a phylogenetic tree was constructed using the ARF sequences from Longya-10, Heiya-14, and pale flax (Appendix A). The analysis showed that most LuARFs had orthologs in both Heiya-14 and pale flax. Another phylogenetic tree was generated using 291 ARF protein sequences, including 7 from dicotyledons (*L. usitatissimum*, *G. hirsutum*, *A. thaliana*, *G. max*, *Vitis vinifera*, *Eucalyptus grandis*, and *Medicago truncatula*) and 3 from monocotyledons (*O. sativa*, *Sorghum bicolor*, and *B. distachyon*) to assess the evolution among the ARFs of diverse species (Figure 2). In this tree, the ARFs were clustered into four clades: clade I, clade II, clade III (clades IIIa, IIIb, and IIIc), and clade IV (clades IVa, IVb, and IVc) (Appendix A). Except for clades IIIb, IVa, and IVb, all other clades contained ARFs from dicotyledons and monocotyledons. Most LuARF proteins clustered closely with the ARFs from *G. hirsutum* and *A. thaliana*. All the 13 LuARF proteins containing the DBD-MR activator-CTD structure, predicted to be transcriptional activators, clustered in clade III, while all the potential transcriptional repressors with the DBD-MR repressor-CTD structure belonged to clade I. In addition, the 14 LuARF proteins without a CTD were grouped as clades I, II, and IV.

### 2.4. Chromosomal Distribution and Synteny Analysis

We further investigated the chromosomal locations of *LuARF* genes (Figure 3). The analysis indicated that 30 *LuARF* genes were distributed unevenly on all chromosomes except for chromosomes 1 and 7. However, we could not map *LuARF31*, *32*, and *33* onto the chromosomes. Chromosome 14, with six members, had the largest number of *LuARFs*. Chromosome 15 contained four *LuARF* genes, chromosomes 2 and 13 contained three genes each, chromosomes 4, 5, 8, 9, and 10 possessed two genes each, and chromosomes 3, 6, 11, and 12 each contained only one gene.

We then analyzed the duplication events to elucidate the mechanism underlying *LuARF* gene family expansion during evolution. The results showed that two gene pairs (*LuARF10*/*LuARF11* and *LuARF22*/*LuARF23*) underwent tandem duplication (TD) and another eighteen gene pairs (*LuARF1*/14, *LuARF2*/*12*, *LuARF3*/*27*, *LuARF4*/*5*, *LuARF4*/*13*, *LuARF5*/*25*, *LuARF5*/*8*, *LuARF8*/*25*, *LuARF6*/*26*, *LuARF7*/*16*, *LuARF9*/*17*, *LuARF10*/*22*, *LuARF15*/*20*, *LuARF15*/*32*, *LuARF20*/*32*, *LuARF32*/*33*, *LuARF19*/*LuARF29* and *LuARF24*/*31*) underwent segmental/whole-genome duplication (SD/WGD) (Figure 3 and Table 2). The genes that experienced SD had one to three syntenic gene pairs in the flax genome (Appendix A). Further, we calculated the nonsynonymous (K_a_) and synonymous (K_s_) values and used the K_a_/K_s_ ratios to reveal the selective forces on the duplicated *LuARF* gene pairs. In general, K_a_/K_s_ > 1 indicates a positive selection, K_a_/K_s_ = 1 indicates a neutral evolution, and K_a_/K_s_ < 1 shows a strong purifying selection [43]. We found that most K_s_ values of these collinear genes varied from 0.05 to 0.23, implying that the duplication events happened close to the recent WGD of flax (K_s_ = 0.13) [40]. The divergence time of these gene pairs was estimated as 4–18 Mya. However, the K_s_ values of *LuARF15/32* and *LuARF20/32* gene pairs were about 0.7, indicating that the duplication happened close to the early WGD event (K_s_ = 0.77), and the divergence time was estimated as 57–60 Mya. In addition, all duplicated gene pairs’ K_a_/K_s_ ratios were less than 0.4, which suggested a strong purifying selection pressure on the *LuARF* genes.

Further, to explore the evolution of *ARF* genes, the collinearity relationships between flax and nine species (six dicots and three monocots) described above were analyzed (Figure 4 and Appendix A). The analysis identified 10, 29, 47, 30, 15, 27, 5, 4, and 6 homologous pairs of *ARFs* between flax and *Arabidopsis*, soybean, cotton, barrel medic, Egrandis, grape, rice, sorghum, and *B. distachyon*, respectively. About 50% of *LuARF* genes had homologous genes in these nine plants, with collinear gene pairs ranging from one to five. In addition, the collinear genes between flax and monocots were far less than that between flax and dicots. A few *LuARFs* had more than three syntenic gene pairs between flax and another two plants (cotton, barrel medic). For example, *LuARF13* and *24* had up to five syntenic gene pairs in cotton, and *LuARF4*, *13*, and *25* had five pairs in barrel medic, indicating that these genes remained highly conserved during evolution. Also, *LuARF4* and *13* had homologous genes in all the selected species, suggesting that these gene pairs might share a common ancestor and existed before the monocotyledon-dicotyledon split. *LuARF25* had orthologs in all species except for *Arabidopsis*, while *LuARF24* had orthologs in seven plants. Except for *LuARF4*, *13*, *24*, and *25*, all other genes lacked orthologs in monocots but had homologs in dicots, implying that most genes might have appeared after the divergence of dicotyledons and monocotyledons.

### 2.5. Phytohormone- and Abiotic Stress-Related Cis-Acting Elements on LuARF Promoter

The *cis*-acting elements in the promoter regions are essential in regulating the transcription of genes. To determine the potential function of *LuARF* genes, the promoter sequences of all *LuARFs* were analyzed using the PlantCARE database [44]. The analysis revealed numerous light, phytohormone, and abiotic stress response elements on *LuARF* promoters and a few basic elements (TATA-box and CAAT-box) (Figure 5 and Appendix A). Further analysis of the hormone- and stress-responsive elements identified five kinds of phytohormone-related elements, including those related to gibberellin (GARE-motif, TATC-box, and P-box), methyl jasmonate (MeJA, CGTCA-motif), salicylic acid (TCA-element and SARE), auxin (TGA-element, AuxRR-core, and TGA-box), and abscisic acid (ABRE). Among them, the gibberellin, MeJA, and abscisic acid-responsive elements were abundant in *LuARF* promoters (44, 63, and 68, respectively). The number of hormone response elements in each *LuARF* ranged from 4 to 12, implying an uneven distribution pattern. *LuARF1*, *2*, *6*, *8*, *10*, *13*, and *32* had all five phytohormone-related elements, while *LuARF20* and *27* contained only two types. *LuARF13* and *22*, *LuARF1*, and *LuARF11* contained up to 4, 6, and 7 elements associated with gibberellin, MeJA, and abscisic acid, respectively. Additionally, three regulatory elements related to environmental stress, the MBS (drought inducibility), LTR (low temperature responsive), and TC-rich repeats (defense- and stress-responsive), were identified in the *LuARF* promoter; 17, 21, and 14 *LuARF* genes contained MBS, LTR, and TC-rich repeats, respectively. Among the 21 genes, *LuARF11* had the most LTR elements.

### 2.6. Tissue-Specific Expression of LuARF Genes

To assess the potential functions of *LuARF* genes, we first analyzed their expression patterns in different tissues (stem and capsule) and cultivars (Longya-10 and Heiya-14) using the previously published transcriptome data (Figure 6a) [40]. The analysis detected the transcripts of 26 *LuARF* genes in both stems and capsules of Longya-10 and Heiya-14, but not that of *LuARF9*, *11*, *23*, and *28*. We found that six *LuARF* genes (*LuARF2*, *4*, *6*, *18*, *25*, and *26*) were highly expressed in capsules, while nine (*LuARF3*, *8*, *13*, *15*, *21*, *22*, *27*, *32*, and *33*) had higher expression levels in stems. Further, we investigated these genes’ expression patterns in flax stem and seeds by qRT-PCR. As shown in Figure 6b, only *LuARF4/13* exhibited differences in the expression patterns between the two cultivars, Longya-14 and Zhangya-2, with different oil content. Subsequent analysis of the correlation between oil content and the expression levels of *LuARF4/13* in seeds at different developmental stages indicated that the expression level in 10 d seeds was positively related to the oil content in Longya-14, while the expression level in 25 d and 40 d seeds was positively associated with the oil content in Zhangya-2 (Appendix A).

To further confirm the importance of the nine genes explicitly expressed in the stem, their expression levels were tested in both whole stems and stem peels at nine positions along the Longya-10 stem (Figure 6c). The nine *LuARF* genes exhibited significantly similar expression patterns. Above the snap point (SA to C), these genes exhibited higher expression in the whole stem than in the stem peel, while the opposite was found at point A. Below the snap point (D to H), *LuARF8* and *LuARF32/33* showed lower expression at position G than C in the stem peel, while the other genes showed obviously higher expression levels at all positions than that at C. The expression patterns of these genes in the whole stem were opposite to that in the stem peel, and the levels diminished from C to D. In general, the nine *LuARFs* exhibited significantly higher expression in the stem peel than in the whole stem at points D to H, with the expression patterns of *LuARF8* and *32/33* at point G and *LuARF21* at point E as the exceptions. *LuARF8* expression at point G was lower in the stem peel than in the whole stem, and *LuARF32/33* expression at point E was lower in the stem peel than in the whole stem.

### 2.7. Expression Profiling of LuARFs under Various Stresses

Finally, to assess the potential functions of *LuARF* genes in response to phytohormones and abiotic stresses, we investigated their expression levels in flax seedlings under IAA, NAA, and GA3 treatments and salinity, drought, cold, and heat stresses. The analysis showed that all *LuARFs* were significantly induced 4 h after exposure to cold and heat stress, and the response of all genes, except *LuARF6/26*, to cold stress was more pronounced than to heat stress (Figure 7a). Under salinity stress (NaCl), all genes were significantly induced at least at one time point after exposure. Most genes exhibited significantly higher expression than the control at 48–72 h after treatment, while *LuARF8* showed higher expression at 9 and 24 h (Figure 7b). After PEG treatment, all *LuARF* genes exhibited an expression peak at 9 h, except for *LuARF7/16*, *LuARF18*, *LuARF19/29*, and *LuARF28* (Figure 7c). Notably, the expression of *LuARF18* and *LuARF28* did not exceed that of the control at any point after PEG treatment. In addition, *LuARF* genes exhibited differential expression patterns under GA3 and IAA treatments (Figure 7d,e). Most genes were significantly upregulated at 3 or 9 h after GA3 and IAA treatments. *LuARF7/16* expression after GA3 treatment was higher than the control at all time points except for 12 h, while *LuARF10*, *LuARF11/23*, and *LuARF32/33* expression levels after IAA exposure were significantly higher than the control at up to six time points. Most *LuARFs* were highly induced at 12 or 24 h after NAA treatment; *LuARF6/26*, *LuARF7/16*, *LuARF10*, *LuARF21*, and *LuARF24/31* were significantly induced at 12, 48, or 72 h, and another three time points (3, 6, 9, or 24 h) after exposure to NAA (Figure 7f). At 48 h recovery, different *LuARFs* exhibited different expression patterns under phytohormone and stress treatments; a few *LuARFs* showed higher expression than the control, while some members showed lower expression than the control.

## 3. Discussion

*ARFs* are pivotal in controlling various biological processes associated with plant growth and development [2]. Although researchers have identified and reported the functions of *ARFs* across multiple plant species, this present work reports the genes in the oilseed and fiber crop, flax. This study comprehensively analyzed the flax *ARF* gene family to elucidate their potential roles in regulating flax growth and response to environmental stimuli.

### 3.1. ARF Genes in Flax

This current research identified 33, 35, and 34 *ARF* members in the genomes of three representative flax cultivars, the conventional oil cultivar Longya-10, the conventional fiber cultivar Heiya-14, and their ancestor pale flax. The three genomes had almost the same number of *ARF* genes, indicating that the *ARF* gene family remained relatively conserved during flax domestication. However, the number of *ARF* genes varied considerably among plant species, such as 33 in flax, 25 in rice [35], 19 in grape [45], 56 in cotton [26], and 40 in *M. truncatula* [46], probably due to the deletion and expansion of genes during the evolutionary process. Further analysis showed extensive variations in gene structure, protein length, MW, and pI of ARFs in the three genomes, implying functional diversity. In addition, all ARF proteins identified in this study were predicted to be located in the nucleus. These observations of LuARFs are consistent with the reports on ARFs of other species [30,32,36,37].

Domain characteristics are crucial for predicting the functions of transcription factors [7]. The present study found that all LuARF proteins had a typical B3 domain (motifs 1, 2, and 6) and an ARF domain (motifs 5, 7, 8, 12, and 14); however, only 19 of the 33 LuARFs had the CTD (motifs 4, 9, and 16), which is not necessary for binding promoter regions, but is necessary for interaction with Aux/IAA proteins to mediate transcriptional levels of downstream genes [7]. The percentage of LuARF without CTD (42.4%) was higher than that in *A. thaliana* (17.39%) [47], *O. sativa* (24%) [35], and *M. truncatula* (37.5%) [46]. Generally, the ARFs without CTD are regulated via interaction with other transcription factors but not Aux/IAA and might be insensitive to auxin and function in an auxin-independent way, such as the *Arabidopsis* AtARF3 [7,9,48]. Therefore, it can be speculated that the 14 LuARFs without the CTD might regulate biological processes in an auxin-independent manner. In addition, we found that 13 out of 19 LuARF members with complete ARF structure contained an MR domain enriched with Q, S, and L, implying their role as transcriptional activators [6,49,50]. The other six LuARFs with CTD and all the LuARFs without CTD might function as transcriptional repressors as their MR domains were enriched with S, L, P, and G, similar to those in other plants [36,46]. These observations showed that the ratio of transcriptional activators and repressors among the LuARF proteins were 0.65, demonstrating that LuARF members evolved distinct functions, as in other plants [34].

Furthermore, the three phylogenetic trees generated in this study demonstrated similar evolutionary relationships for the LuARFs, indicating the accuracy of the phylogeny analysis. All clades (I, II, IIIa, IIIc, and IVc) contained ARFs from both dicots and monocots, suggesting relatively conserved ARF functions among the species. However, clade IIIb and IVb were dicot-specific subgroups and clade IVa only had one monocot ARF member, suggesting specific roles for these ARFs. Our study also found that the LuARFs within the same clade had similar exon–intron structures and domain composition, similar to those of other species [32,36,46], indicating functional redundancy. We then found 4 *LuARF* genes (2 gene pairs) experienced TD events, while 27 (81.8%, 18 gene pairs) experienced SD events. Studies have demonstrated the significance of SD/WGD and TD in gene expansion and functional differentiation of multiple gene families [51,52]. Generally, SD results in functional redundancy, while TD results in novel functions, which help the species adapt to the fast-changing environment [53]. Subsequent K_s_ analysis implied that the most recent WGD event was the leading cause of the expansion of the *LuARF* gene family [40]. In *S. italica*, the WGD event caused the expansion of the *ARF* gene family [36]. Thus, our observations suggest that the continual duplication of *LuARF* genes might have contributed to function specificity or redundancy. Additionally, the K_a_/K_s_ ratios of all the duplicated gene pairs were less than 0.4, indicating a purifying selection force on *LuARF* genes (Table 2). Research has proven that *ARF* members demonstrate relatively conserved homologous relationships among species [54]. In the present study, comparing the *LuARFs* with the genes of six dicots and three monocots revealed that about half of *LuARFs* had homologous genes in the analyzed species. In addition, monocots had fewer collinear gene pairs than the dicots, implying that *ARF* genes have experienced a wide range of evolutionary and replication events after the split of monocot and dicot plants, as in *Prunus avium* [32], probably leading to the unique features of monocot and dicot plants.

### 3.2. LuARFs’ Possible Role in Regulating the Development of Organs and Responses to Exogenous Stimuli

Spatial expression profiling of genes helps elucidate their functions. In this study, we generated a heatmap based on the transcriptome data of capsules and stems and analyzed the expression of a few selected *LuARF* genes (specifically expressed) in seeds and stems at different development stages. *LuARF4* showed higher expression in capsules and exhibited different expression patterns in cultivars with different oil content (Figure 6b). Moreover, its expression level was positively related to oil content (Appendix A). Thus, *LuARF4* was speculated to regulate genes related to fatty acid synthesis. In flax, the specification and elongation of fiber occur above the snap point, while the thickening of the fiber cell wall occurs below [55]. Studies have indicated that genes with increased expression levels below the snap point in stem peels but decreased levels in the whole stem and whose expression levels in stem peels at least were as high as those in whole stems at equivalent positions might be involved in the strengthening of fiber cell wall when the fiber cells stop elongating [56,57]. In *Arabidopsis*, *CESA4* and *CESA8* with high expression in phloem peel have been associated with cellulose microfibril synthesis in the secondary cell wall [58]. In flax, *LuPME1* showed increased expression below the snap point in stem peels, with an expression 50 times higher than in the whole stem at the same position [56]. These observations suggested its role in the thickening of the cell wall when the cells stop elongating. In this study, below the snap point, nine *LuARF* genes showed higher expression in stem peel than in the whole stem at the same position. Thus, based on the earlier reports and the current findings, *LuARF3*, *8*, *13*, *15*, *21*, *22*, *27*, *32*, and *33* could be associated with the thickening of the flax fiber cell wall.

The ARF proteins are known to play key roles in the auxin signaling pathway, and their activities are regulated by auxin concentrations [59]. Additionally, studies have identified that *ARF* genes, with *cis*-acting elements related to hormone or abiotic stresses, regulate plant responses to auxin or abiotic stress [30,34,46]. The present study identified 6 to 16 *cis*-acting elements associated with the response to hormones and environmental stimuli in the *LuARF* genes, indicating their vital role in hormone and stress responses. Therefore, we further analyzed the effect of environmental stresses and exogenous hormones on *LuARF* expression to elucidate their roles in stress and phytohormone responses. Our experiments demonstrated that all *LuARF* genes were significantly upregulated after exposure to IAA, NAA, and GA3, probably because exogenous hormones affect in vivo auxin homeostasis and, in turn, the expression of these genes [59]. These observations are consistent with the findings on the response of *ARF* genes in other species to IAA, NAA, and GA3 treatments [30,36,60]. We also found that, under the same treatment, different *LuARF* members showed different expression patterns, indicating differences in their functions in the auxin signal transduction pathway. The expression patterns of *LuARF* genes were also different under the three hormone treatments, which implied distinct regulatory modes of *LuARF* genes when subjected to different treatments. In addition, all *LuARF* genes were significantly upregulated under cold and heat stress, consistent with the observations in *B. distachyon* [60]. In addition, all *LuARF* members were responsive to NaCl and PEG treatment, consistent with the *ARF* genes of tomato under salt and drought stress [61]. For example, *IbARF5* of sweet potato significantly improved salt and drought tolerance of *Arabidopsis*, and this response was associated with ABA biosynthesis [62]. Previous studies have also indicated that the *ARF* genes might resist heat stress [63] and respond to salt and drought stresses [36] through ABA-mediated signaling pathways. Therefore, it could be speculated that *LuARF* genes regulate osmotic stress response ABA-dependently.

## 4. Materials and Methods

### 4.1. Plant Growth and Abiotic Stress Treatments

The flax genotypes Longya-10, Longya-14, Longya-15, and Zhangya-2 were chosen for this study. Longya-14, Zhangya-2, and Longya-10 were cultivated in the experimental field of Gansu Academy of Agricultural Sciences in Lanzhou, Gansu Province (103°40′ E, 34° N). The seeds collected from Longya-14 and Zhangya-2 plants at 5, 10, 15, 20, 25, 30, 40, and 50 days post anthesis and the stem tissues collected from Longya-10 plants (52~54 cm) at about 5 weeks post-emergence were used to determine the expression patterns of *LuARF* genes. Here, according to the sampling method described by Pinzon-Latorre and Deyholos [56], the whole stem (1-cm-long sections) or the stem peel from Longya-10 plants was collected from nine positions based on the fiber development stage as follows: 0 to 1 cm (shoot apex, SA), 1 to 2 cm (A), 2 to 3 cm (B), 3 to 4 cm (C), 10 to 11 cm (D), 20 to 21 cm (E), 30 to 31 cm (F), 40 to 41 cm (G), and 50 to 51 cm (H).

Then, to investigate the expression patterns of *LuARF* genes under various phytohormone and abiotic stress treatments, seeds of Longya-15 were allowed to germinate in Petri dishes at 25 °C under 16 h:8 h light/dark conditions in a climate chamber. After 10 days of growth, uniformly grown seedlings were selected, transferred to 1/2 MS liquid medium, and cultivated for 3 days under the same conditions. Then, the medium was replaced with fresh medium containing 150 mM NaCl and 20% PEG to induce salinity and drought treatment, respectively, and 15 µM indole-3-acetic acid (IAA) or 5 µM naphthalene acetic acid (NAA) or 5 µM gibberellic acid (GA3) for the phytohormone treatment. The seedlings were subjected to 42 °C and 4 °C to induce heat and cold stress, respectively. The plants used for treatments were all from a single sowing. For each treatment, three biological replicates were set up. The leaves were collected from seedlings under heat and cold stress after 3 h of exposure and from drought, salinity, and phytohormone treatment at 3, 6, 9, 12, 24, 48, and 72 h under stress and 48 h after subsequent growth in 1/2 MS liquid medium without supplement. The leaves from the control seedlings were collected at 0 h and all other time points. For each treatment, samples were collected from three plants per replicate at each time point. All samples were frozen in liquid nitrogen immediately after collection and stored at −80 °C for RNA extraction.

### 4.2. Identification of ARF Genes

The genome sequences of *L. usitatissimum* (Longya-10 and Heiya-14) and *Linum bienne* (pale flax) were obtained from NCBI (https://www.ncbi.nlm.nih.gov/, accessed on 13 December 2022), and their annotation files were downloaded from Figshare (https://figshare.com/, accessed on 13 December 2022). The ARF protein sequences of *G. hirsutum*, *A. thaliana*, *G. max*, *V. vinifera*, *E. grandis*, *M.truncatula*, *O. sativa*, *S. bicolor*, and *B. distachyon* were obtained from Phytozome v13 (https://phytozome-next.jgi.doe.gov/, accessed on 13 December 2022) and used as queries to identify the *ARF* genes in Longya-10, Heiya-14, and pale flax using BLASTP analysis (e-value of 1 × 10^−5^) [41]. All candidate ARF proteins were further examined for the presence and integrity of the conserved domains [Pfam 06507: ARF (AUX_RESP); Pfam 02362: B3 DNA binding domain (B3)] using the Pfam (http://pfam-legacy.xfam.org/search#tabview=tab1, accessed on 15 December 2022) and the NCBI Conserved Domain Databases (CDD, https://www.ncbi.nlm.nih.gov/cdd/, accessed on 15 December 2022). Finally, the non-redundant *LuARF* genes were obtained using one gene model per locus. The MW and pI of all LuARF proteins were computed using the ExPASy ProtoParam (http://web.wxoasy.org/protparam/, accessed on 18 December 2022) tool [64] and the subcellular localization was predicted using Plant-mPloc (http://www.csbio.sjtu.edu.cn/ bioinf/plant-multi/, accessed on 18 December 2022) [65]. MEME program was used to identify and analyze the conserved motifs of the LuARF proteins using default parameters (http://alternate.meme-suite.org/tools/meme, accessed on 24 December 2022) [66].

### 4.3. Chromosomal Mapping and Collinearity Analysis

The chromosomal location of *LuARF* genes was obtained from the annotation files downloaded from Figshare, and the chromosome distribution map was drawn using Map Gene 2 Chromosome (MG2C, http://mg2c.iask.in/mg2c_v2.0/, accessed on 4 January 2023). The MCScanX package was adopted to analyze the *LuARF* gene duplication events using default parameters [67], and the TBtools software (version 1.9876.0.0) was used to assess and visualize the collinearity among the *ARFs* [68]. The K_a_ and K_s_ substitution rates per site of the duplicated gene pairs were calculated using DnaSP software (version 6.0) [69], and the K_a_/K_s_ ratio was used to assess the mode and strength of the selective pressure [43]. For each gene pair, the divergence time (million years ago, Mya) was also calculated using the following formula [70]:Mya = Ks/(2 × 6.1 × 10^−9^) × 10^−6^.

### 4.4. Gene Structure and Promoter Analysis

The exon–intron organization of the *LuARF* gene family members was determined based on the genome and coding sequences and visualized using the GSDS (http://gsds.cbi.pku.edu.cn/, accessed on 4 January 2023) [42]. The 2000 bp region upstream of each *LuARF* start codon was extracted from the Longya-10 genome, and the *cis*-acting elements were determined using the PlantCARE database (http://bioinformatics.psb.ugent.be/webtools/plantcare/html/, accessed on 4 January 2023) [44].

### 4.5. Phylogenetic Analysis

The full-length amino acid sequences of the ARFs of *L. usitatissimum*, *V. vinifera*, *G. max*, *G. hirsutum*, *A. thaliana*, *E. grandis*, *M. truncatula*, *S. bicolor*, *B. distachyon*, and *O. sativa* were used to construct a phylogenetic tree. First, a multiple sequence alignment was performed with ClustalX in MEGA (6.0) using default parameters. Then, a phylogenetic tree was built using the alignment files based on the neighbor-joining (NJ) method with 1000 bootstrap replicates [71].

### 4.6. Gene Expression Analysis

Previously published transcriptome data on the capsule and stem of Longya-10 and Heiya-14 (PRJNA505721) were initially used to determine the expression patterns of *LuARFs* in different tissues and genetic backgrounds [40], and the TBtools software (version 1.9876.0.0) was used to draw a heat map based on these expression patterns. Then, the *LuARF* genes with high expression levels in the capsule or stem were further analyzed using qRT-PCR. The expression profiles of *LuARFs* under various treatments were also determined using qRT-PCR. Total RNA was extracted by using EZgene^TM^ Plant Easy Spin RNA Miniprep Kit following the manufacturer’ s instructions (BIOMIGA, San Diego, CA, USA), and cDNA synthesis was performed by using PrimeScript^TM^ RT reagent Kit with gDNA Eraser (Takara, San Jose, CA, USA). The specific primer pairs used for qRT-PCR were designed by Primer Premier 5.0 (PREMIER Biosoft International, Palo Alto, CA, USA) and listed in Appendix A. qRT-PCR was performed by using TB Green^TM^ Premix Ex Taq^TM^ II (Takara, San Jose, CA, USA) and conducted on Eco Real-Time PCR System (Illumine). The reaction condition was as follows: 2 min at 50 °C, 10 min at 95 °C, with 40 cycles of 15 s at 95 °C, 15 s at 60 °C, and 15 s at 72 °C, followed by 10 s at 95 °C, 1 min at 65 °C, and 1 s at 97 °C. Each sample had three replicates. *GAPDH* was used as an internal reference gene [72], and the relative expression levels of the *LuARF* genes were calculated following the 2^−ΔΔCt^ method [73]. The coding sequences of the *LuARF1/14*, *LuARF2/12*, *LuARF3/27*, *LuARF4/13*, *LuARF5/25*, *LuARF6/26*, *LuARF7/16*, *LuARF9/17*, *LuARF11/23*, *LuARF15/20*, *LuARF19/29/30*, *LuARF24/31*, and *LuARF32/33* gene pairs were extremely similar, and therefore, it was impossible to design primers specific for each gene of the pair. Thus, only one primer pair was designed to amplify the genes of each pair.

## 5. Conclusions

In conclusion, the present study identified 33 *LuARF* genes in Longya-10 (oil-use flax variety), and their gene structure, protein features, phylogeny, promoter elements, and expression patterns were analyzed. The results showed that *LuARFs* can function as either transcriptional activators or repressors, and the evolution and expansion of *LuARFs* were mainly driven by WGD events. Moreover, a positive association between *LuARF* gene expression and oil content was observed. All these results revealed the important roles of *LuARFs* in evolution, organ development, and responses to stress and phytohormone.

## Figures and Tables

**Figure 1 ijms-24-11006-f001:**
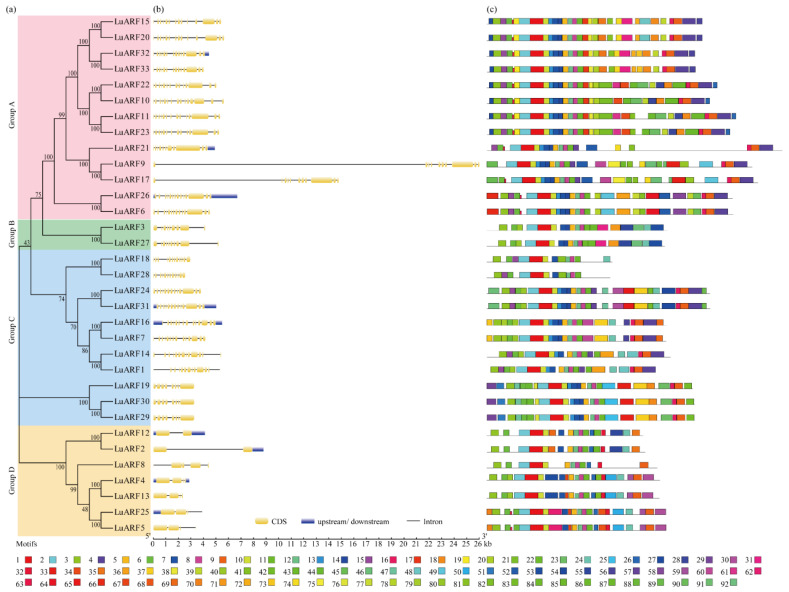
Evolutionary relationship, gene structure, and conserved motifs of LuARFs. (**a**) Phylogenetic tree of LuARF proteins conducted using the NJ method through MEGA 6.0. (**b**) Exon–intron organization of *LuARF* genes. The exons, introns, and upstream/downstream are represented by yellow filled boxes, black lines, and blue filled boxes, respectively. (**c**) Motif patterns of LuARF proteins detected using MEME. Different colors indicate different motifs.

**Figure 2 ijms-24-11006-f002:**
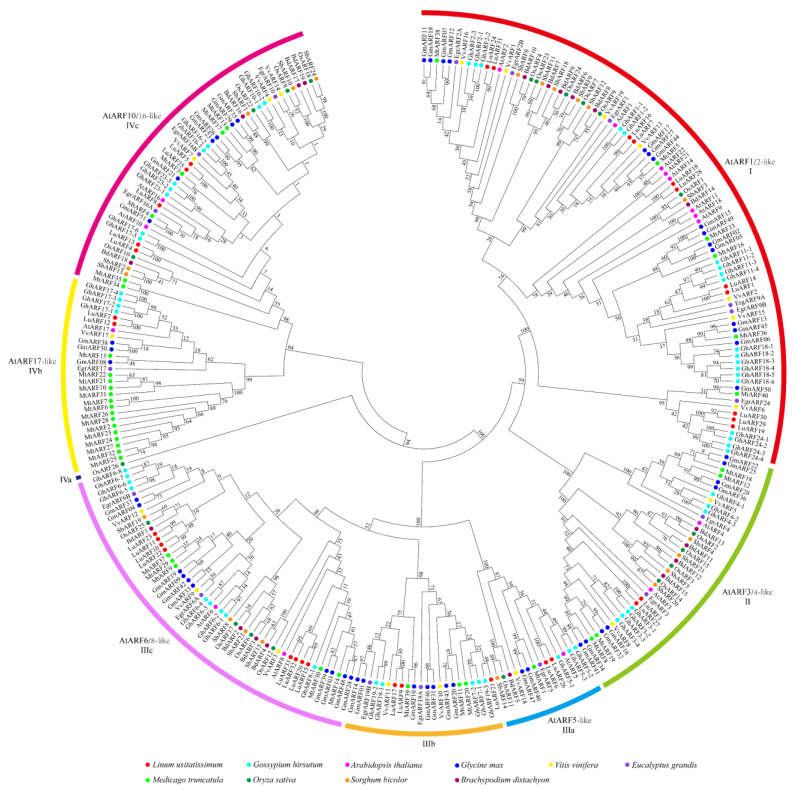
Phylogenetic tree of ARF proteins from flax, *Arabidopsis*, soybean, cotton, barrel medic, Egrandis, grape, rice, sorghum, and *B. distachyon*. All ARFs are divided into four main clades, and clades III and IV were further divided into three subgroups, each indicated by a different color. ARF proteins obtained from different species were marked by circles filled with different colors.

**Figure 3 ijms-24-11006-f003:**
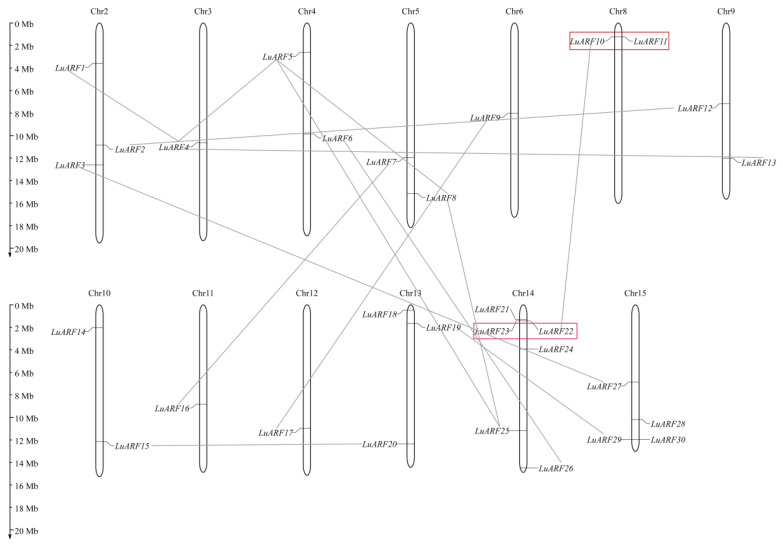
Chromosomal distribution of *LuARF* genes. Red rectangles and gray lines show the gene pairs underwent tandem and segmental duplication, respectively.

**Figure 4 ijms-24-11006-f004:**
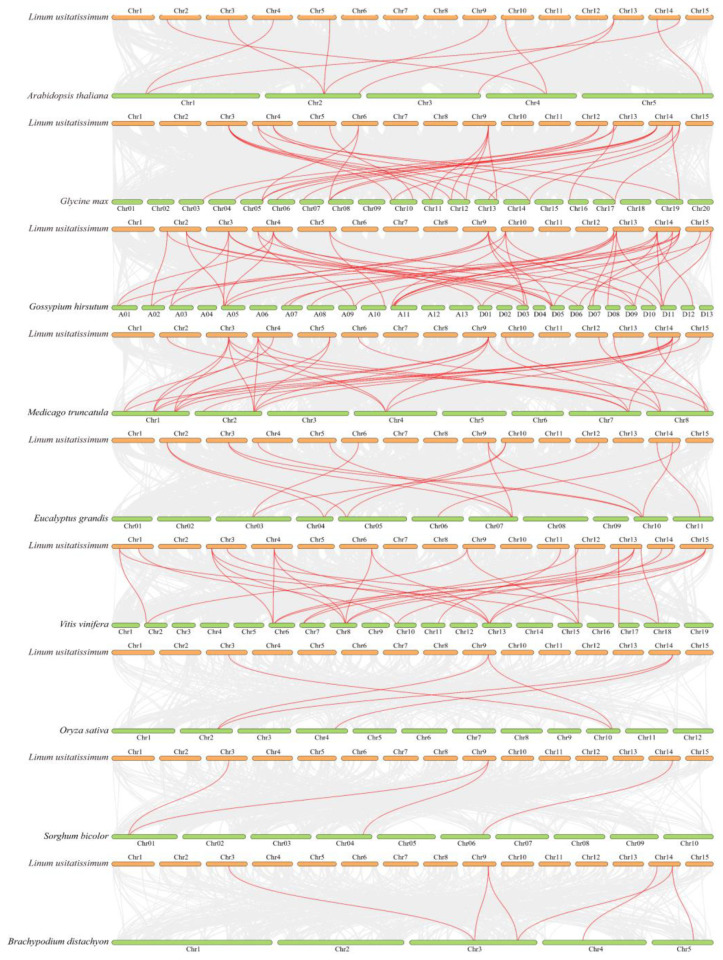
Collinearity analyses of *ARF* genes between flax and other nine species (*Arabidopsis*, soybean, cotton, barrel medic, Egrandis, grape, rice, sorghum, and *B. distachyon*). Gray lines indicate the syntenic blocks between flax and other plant genomes, whereas the red lines represent the collinear *ARF* gene pairs.

**Figure 5 ijms-24-11006-f005:**
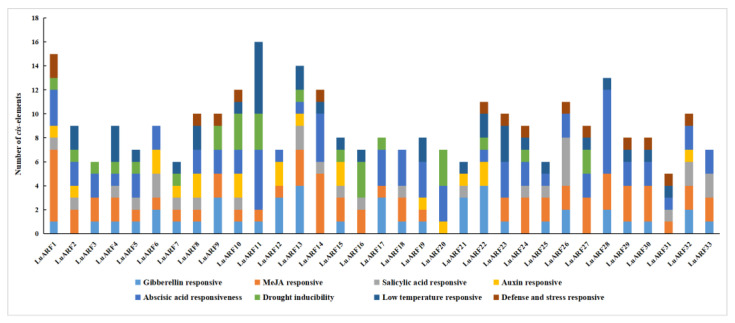
Promoter analysis of *LuARF* genes. The number of different *cis* elements are presented in the form of column charts, which are indicated by different colors.

**Figure 6 ijms-24-11006-f006:**
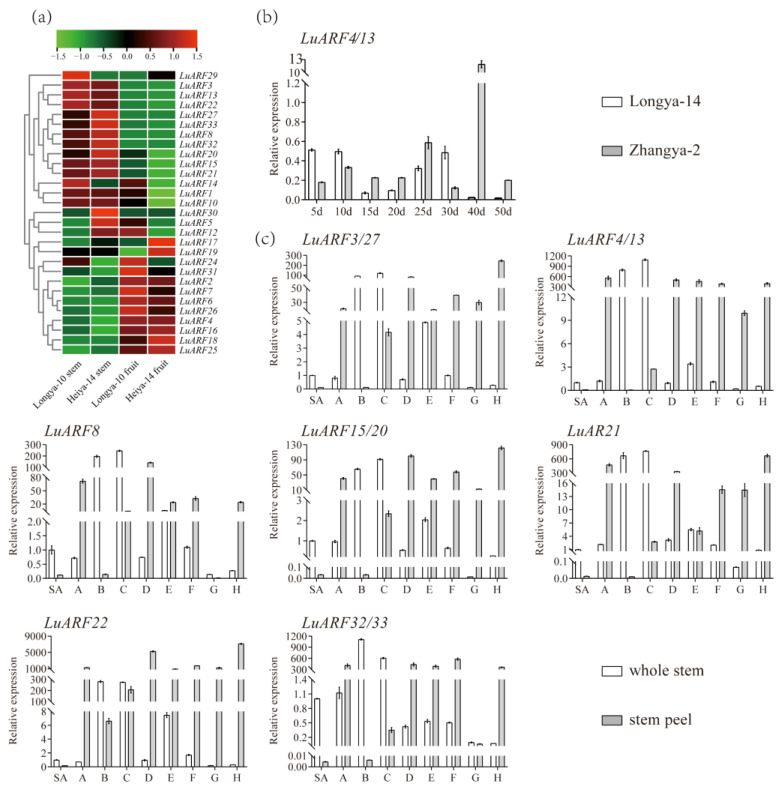
Expression patterns of *LuARF* genes in different tissues of flax. (**a**) Expression analysis of *LuARFs* in different tissues and varieties using transcriptome data. (**b**) Relative expression levels of *LuARF4/13* in seeds of two flax varieties at different developmental stages. (**c**) Relative expression levels of *LuARF* genes in whole stem and stem peel. SA, A, to H represent the whole stem or the stem peel collected from 0 to 1 cm, 1 to 2 cm, 2 to 3 cm, 3 to 4 cm, 10 to 11 cm, 20 to 21 cm, 30 to 31 cm, 40 to 41 cm, and 50 to 51 cm of Longya-10 plants based on the fiber development stage, respectively. All the results were based on three biological replicates. Error bars indicate ±SD for three biological replicates.

**Figure 7 ijms-24-11006-f007:**
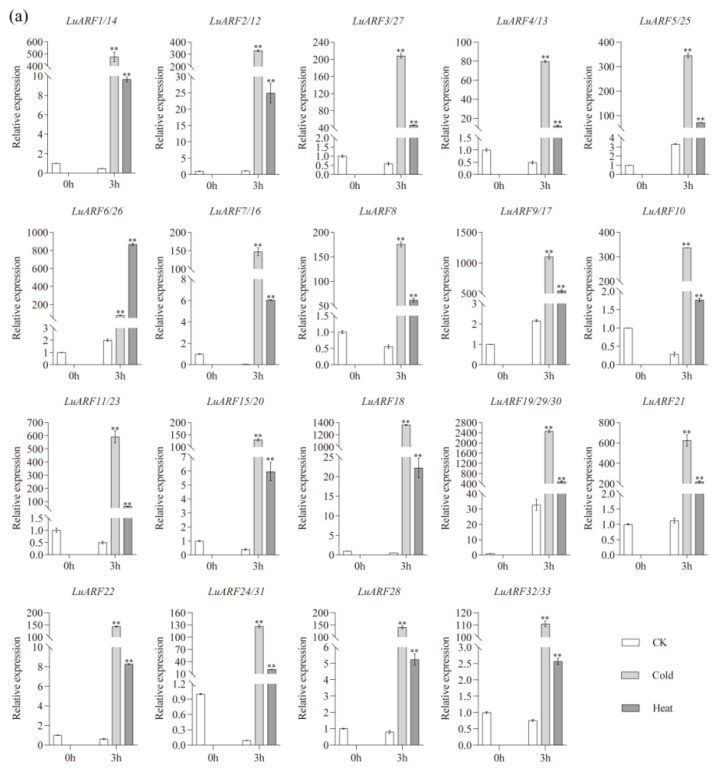
Expression profiles of *LuARF* genes under (**a**) cold and heat, (**b**) NaCl, (**c**) PEG, (**d**) GA3, (**e**) IAA, and (**f**) NAA treatments. CK indicates samples collected from the control seedlings. All the results were based on three biological replicates. Error bars indicate ±SD for three biological replicates. * and ** represents the significant differences at the 0.05 and 0.01 level, respectively.

**Table 1 ijms-24-11006-t001:** Basic characteristics of the *LuARF* genes in flax (Longya-10).

Gene Name	Locus ID	Chromosome Location	Length (aa)	MW (KDa)	pI	Exons	Subcellular Localization
*LuARF1*	L.us.o.m.scaffold38.111	Chr2: 4056063-4061396(−)	654	73.74	6.43	13	Nucleus
*LuARF2*	L.us.o.m.scaffold177.42	Chr2:12271477-12280264(+)	602	65.95	6.05	2	Nucleus
*LuARF3*	L.us.o.m.scaffold33.48	Chr2:14271938-14276080(+)	688	75.21	6.98	9	Nucleus
*LuARF4*	L.us.o.m.scaffold20.385	Chr3:12023683-12026552(+)	659	72.31	7.85	4	Nucleus
*LuARF5*	L.us.o.m.scaffold183.107	Chr4:2988619-2992012(+)	682	75.76	6.98	4	Nucleus
*LuARF6*	L.us.o.m.scaffold336.4	Chr4:11172052-11176570(−)	937	102.77	5.60	14	Nucleus
*LuARF7*	L.us.o.m.scaffold2.53	Chr5:13559053-13563212(−)	683	75.98	5.81	14	Nucleus
*LuARF8*	L.us.o.m.scaffold255.60	Chr5:17150878-17155311(−)	649	72.37	8.09	5	Nucleus
*LuARF9*	L.us.o.m.scaffold53.226	Chr6:9052662-9078679(+)	1010	112.57	6.67	10	Nucleus
*LuARF10*	L.us.o.m.scaffold82.237	Chr8:1405577-1411201(+)	847	93.84	6.04	15	Nucleus
*LuARF11*	L.us.o.m.scaffold82.236	Chr8:1413136-1418471(+)	948	105.43	6.39	14	Nucleus
*LuARF12*	L.us.o.m.scaffold33.141	Chr9:8160867-8164989(−)	595	65.11	6.76	2	Nucleus
*LuARF13*	L.us.o.m.scaffold131.61	Chr9:13620512-13622865(−)	657	72.43	6.81	4	Nucleus
*LuARF14*	L.us.o.m.scaffold9.422	Chr10:2282901-2288310(+)	698	78.45	6.78	13	Nucleus
*LuARF15*	L.us.o.m.scaffold199.64	Chr10:13744807-13750216(+)	820	91.32	6.16	14	Nucleus
*LuARF16*	L.us.o.m.scaffold5.303	Chr11:9965217-9970709(−)	684	76.11	5.72	14	Nucleus
*LuARF17*	L.us.o.m.scaffold3.144	Chr12:12422846-12437639(+)	1032	114.79	6.28	11	Nucleus
*LuARF18*	L.us.o.m.scaffold75.41	Chr13:524613-527571(−)	478	52.82	8.49	12	Nucleus
*LuARF19*	L.us.o.m.scaffold71.2	Chr13:1846189-1849447(−)	784	87.26	6.33	8	Nucleus
*LuARF20*	L.us.o.m.scaffold83.44	Chr13:14011405-14017053(−)	820	91.39	6.12	14	Nucleus
*LuARF21*	L.us.o.m.scaffold48.39	Chr14:1479346-1484254(−)	1123	125.34	6.31	11	Nucleus
*LuARF22*	L.us.o.m.scaffold48.78	Chr14:1639106-1644136(+)	876	97.13	5.87	14	Nucleus
*LuARF23*	L.us.o.m.scaffold48.79	Chr14:1646009-1651254(+)	924	102.67	6.23	14	Nucleus
*LuARF24*	L.us.o.m.scaffold56.318	Chr14:4404281-4408082(−)	849	94.30	6.31	13	Nucleus
*LuARF25*	L.us.o.m.scaffold168.120	Chr14:12632364-12636289(+)	682	75.57	6.55	4	Nucleus
*LuARF26*	L.us.o.m.scaffold122.81	Chr14:16412628-16419332(+)	936	102.87	5.45	14	Nucleus
*LuARF27*	L.us.o.m.scaffold107.81	Chr15:7797623-7802834−)	677	74.04	6.88	10	Nucleus
*LuARF28*	L.us.o.m.scaffold0.555	Chr15:11542664-11545214(−)	470	52.81	8.59	12	Nucleus
*LuARF29*	L.us.o.m.scaffold239.1	Chr15: 13529432-13532699(−)	792	88.22	6.45	8	Nucleus
*LuARF30*	L.us.o.m.scaffold156.62	Chr15: 13548791-13552058(−)	792	88.22	6.38	8	Nucleus
*LuARF31*	L.us.o.m.scaffold81.17		850	94.28	6.45	13	Nucleus
*LuARF32*	L.us.o.m.scaffold165.136		791	87.68	5.73	14	Nucleus
*LuARF33*	L.us.o.m.scaffold434.2		792	87.79	5.86	14	Nucleus

Note: *LuARF31*, *LuARF32*, and *LuARF33* were not mapped on any chromosome.

**Table 2 ijms-24-11006-t002:** Duplicate events, selection pressure, and divergence time of *LuARF* genes.

Gene Pairs	Duplication Event	K_a_	K_s_	K_a_/K_s_	Selection Type	Divergence Time (Mya)
*LuARF10*/*LuARF11*	TD	0.0495	0.3311	0.1495	Purifying selection	27.1393
*LuARF22*/*LuARF23*	TD	0.0566	0.3028	0.1869	Purifying selection	24.8197
*LuARF1*/*LuARF14*	SD	0.0224	0.1009	0.2220	Purifying selection	8.2705
*LuARF2*/*LuARF12*	SD	0.0375	0.1097	0.3418	Purifying selection	8.9918
*LuARF3*/*LuARF27*	SD	0.0319	0.1731	0.1843	Purifying selection	14.1885
*LuARF4*/*LuARF5*	SD	0.3173	2.7505	0.1154	Purifying selection	225.4508
*LuARF4*/*LuARF13*	SD	0.0404	0.1177	0.3432	Purifying selection	9.6475
*LuARF5*/*LuARF25*	SD	0.0196	0.1073	0.1827	Purifying selection	8.7951
*LuARF5*/*LuARF8*	SD	0.2794	1.9360	0.1443	Purifying selection	158.6885
*LuARF8*/*LuARF25*	SD	0.2789	2.4493	0.1139	Purifying selection	200.7623
*LuARF6*/*LuARF26*	SD	0.0090	0.0735	0.1224	Purifying selection	6.0246
*LuARF7*/*LuARF16*	SD	0.0159	0.0556	0.2860	Purifying selection	4.5574
*LuARF9*/*LuARF17*	SD	0.0158	0.1057	0.1495	Purifying selection	8.6639
*LuARF10*/*LuARF22*	SD	0.0192	0.1612	0.1191	Purifying selection	13.2131
*LuARF15*/*LuARF20*	SD	0.0372	0.1141	0.3260	Purifying selection	9.3525
*LuARF15*/*LuARF32*	SD	0.1029	0.6960	0.1478	Purifying selection	57.0492
*LuARF20*/*LuARF32*	SD	0.1048	0.7268	0.1442	Purifying selection	59.5738
*LuARF32*/*LuARF33*	SD	0.0062	0.2226	0.0279	Purifying selection	18.2459
*LuARF19*/*LuARF29*	SD	0.0250	0.1660	0.1506	Purifying selection	13.6066
*LuARF24*/*LuARF31*	SD	0.0067	0.0681	0.0984	Purifying selection	5.5820

Note: TD indicates tandem duplication; SD indicates segmental duplication.

## Data Availability

All data are reported in the article and the Appendix A.

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
