# Peer review of "Genome-Wide Identification and Expression Analysis of Auxin Response Factor Gene Family in Linum usitatissimum"

_ijms, 2023, doi:10.3390/ijms241311006_

Round 1

Reviewer 1 Report

Overall, this is a soundly conducted study that is novel and will be of interest to those studying flax and ARF transcription factors. The author’s work focuses on ARF transcription factors of Lu and important oilseed crops. ARFs are known to play an important role in plant growth and development and have been identified in multiple species. Nevertheless, no studies have been conducted for L. usitatissimum. The author’s use of the current literature and evolutionary biology to identify these ARFs within flax, and further characterize them according to their domains. This research is a base for future studies and crop improvement by tuning ARFs expression according to their roles in planta

This manuscript could be greatly improved by including citations to more recent advances in the molecular knowledge of ARF function as well as methods, increasing figure resolution, and providing more explicit figure legends and reproducible methods. Detailed line-by-line references to proposed changes are below. 

Line-by-line comments

11 - “the critical components” is a bit too strong in my mind. I would rephrase as “are critical components … , and are involved …

44 - The canonical TGTCTC AuxRE has been questioned in recent years (https://doi.org/10.1073/pnas.2009554117) and using TGTCNN and/or a mention of the preference of some ARFs for TGTCGG may be worthwhile.

48 - The use of “C-terminal dimerization domain” here and throughout is a bit misleading as these PB1 (Phox-Bem1) C-terminal domains can actually form oligomers. https://doi.org/10.1073/pnas.1400074111 and similar citations.

63 - In this paragraph it is a bit awkward to switch between protein (non-italicized) and gene (italicized) nomenclature. I would change all of these to genes and italicize them.

98 - Are the query sequences or locus IDs available as a supplemental table? This would be a nice addition, if not. 

98 - BLASTP should be cited

100 – Number of ARFs (35 and 34) does not match Table S1. If I am understanding it correctly. Maybe all duplicated IDs are included? 

104 – Specify Molecular Weight (MW) here instead of 108. Same applies to the isoelectric point (pI).

116/117 - “divided” is repeated. I also believe that the sentence containing the word could be improved.

119 - GSDS acronym is not defined and should also be cited.

128 - Figure 1 resolution could be increased, actually all figures could be of higher resolution as they are fuzzy when zoomed in to improve visibility.

161 - typo in Ivb should be IVb.

164 - Scientific names of the majority of plant species are not italicized. Not only in this line but elsewhere in the manuscript. Same applies to gene names such as LuARF, unless they are being represented as protein sequences.

169 - Fig2 Is clade IIIC AtARF7/19-like?

193 - Ka/Ks, should have a and s as subscripts. 

194 - The novice reader will appreciate a brief description and reference for the Ka/Ks statistic, and what typical values indicate.

200 - “Besides” here is a bit confusing as it can mean “except”. I think “In addition” would be a better transitional phrasing.
208 - Again italics are missing for many scientific names and gene names.

229 - PlantCARE database should be cited.

283 - SA, A - H need to be defined in the legend. Also, replication information and what error bars represent needs to be added.

308 - What does CK stand for? This should be added to the legend. 

322 - Metabolism here is a bit too broad. The discussion should be more specific about your results. You studied ARFs expression in relation to oil content and development. 

334 - This sentence should read “observations of…”

339 - The CTD is not necessary for binding promoters, but is necessary for interaction with Aux/IAA proteins to mediate transcriptional repression through TPL corepressors.

349 - “Might probably” is redundant. Use just one or the other.

370 - Refer back to the relevant table.

384–386 - Again referring to figures/tables would be useful.

370–371 - This is vague: I am not  sure if this sentence conveys a lot of information to me.

392-400 - What is the relationship of the genes mentioned and the LuARFs under investigation.

455 - It should be clarified if these 9 plants were all from a single sowing and experimental replication, or if there was any replication of the whole experiment.

510 - Including the methods used for RNA prep and qPCR in this paper as opposed to citing another would be preferable. 

The english language here is of high quality for the most part with only a few exceptions of atypical word choices as identified in our line-by-line comments.

Reviewer 2 Report

Authors has done a big work, presented comprehensive results about the structure of auxin response factor genes in Linum usitatissimum. During the study authors compared the structure of ARF genes with some crop and model species. The conclusion that ARF genes are multiple and differ among species is a proof that many copies of genes protect basic processes in an organism. At the same time authors found out different expression response of ARF genes to certain abiotic stresses and hormonal treatments.

I have only few comments:

Line 71 – it seems it should be “lateral root development”

Lines 84-86 – give a reference to an article about the release of flax genome

Lines 157-164 – check italic for species names. In addition, Arabidopsis is a genus name, so it is better to write it with italic, even if you use it without species name.

Line – 441 – I guess it should be Longya-14. If not, it should be noted in line 430.  Give a detailed protocol of seed germination, sterilization if used. What is “after ten days, uniformly grown seedling sere selected”? Did germination take 10 days, or these 10 days is a period of growth? If it is a period of growth, conditions should be given.
